# Has China’s Low-Carbon City Construction Enhanced the Green Utilization Efficiency of Urban Land?

**DOI:** 10.3390/ijerph19169844

**Published:** 2022-08-10

**Authors:** Bing Kuang, Jinjin Liu, Xiangyu Fan

**Affiliations:** 1College of Public Administration, Central China Normal University, Wuhan 430079, China; 2Institute of Nature Resources Governance, Central China Normal University, Wuhan 430079, China; 3College of Public Administration, Huazhong University of Science and Technology, Wuhan 430074, China

**Keywords:** low-carbon city pilot, the green utilization efficiency of urban land, propensity score matching difference-in-differences method, policy evaluation

## Abstract

China has implemented the low-carbon city pilot (LCCP) policy in the hopes of efficiently limiting carbon emission intensity to combat global warming and promote green economic growth. Urban land utilization, the second-largest source of carbon emissions, is key to the LCCP policy being able to have the desired effect, which has attracted widespread attention. Based on the panel data from prefecture-level cities in China from 2006 to 2019, this study used the propensity score matching difference-in-differences method (PSM-DID) to examine the impacts of LCCP policy on green utilization efficiency of urban land (GUEUL). The results reveal that LCCP policy has a beneficial impact on GUEUL and can effectively boost the future possibilities of green and low-carbon city development. Due to variances in regional economic and resource endowment level, the impacts of LCCP are different. The pilot has pushed GUEUL in the eastern region, western region, and growing resource-based cities, but has failed to improve GUEUL in other regions. Policymakers should adhere to the long-term sustainability of the LCCP policy and adopt differentiated action strategies to promote GUEUL when implementing it in different regions.

## 1. Introduction

Global carbon emissions are estimated to have grown by 25–30% in the last 200 years. How to reduce carbon emissions effectively has gained increasing attention worldwide [1,2]. China, the largest carbon emitter and the second-largest economy in the world, has committed to achieving peak carbon by 2030 and carbon neutrality by 2060. China has also adopted strict measures to address the current challenges, among which the “Low-Carbon City Pilot” (LCCP) policy is a representative one. During the implementation of this policy, not only the total amount and intensity of carbon emissions are controlled, but also green development is promoted. Green development is a rational consideration of the relationship between economic development and environmental ecology. Further, low carbon and green are emphasized in various fields, such as urbanization development, industrial structure or distribution, and residents’ life [3]. These will eventually be mapped to the sustainable development and utilization of land resources. The traditional pattern of urban land utilization with high intensity and high pollution has been unable to meet the requirements of low-carbon and green development. Improving the green utilization efficiency of urban land (GUEUL) has, therefore, become an important target in low-carbon development.

A rising number of research has explored the GUEUL and its influencing factors. Regarding the efficiency evaluation of urban land utilization, existing studies are often based on the DEA model from an input and output perspective and integrate undesirable outputs, such as carbon emissions resulting from urban construction land utilization, urban wastewater, and urban waste gas emissions [4,5,6]. In addition, indicator evaluation is another representative method, including two types, comprehensive indicators and single indicator. Ustaoglu E. and Aydınoglu [7] evaluated the suitability of urban construction land through multiple indexes, including geographical qualities, accessibility, conditions in built-up areas, urban greening, and living welfare. He et al. [8] used the added value of the secondary and tertiary industries per square kilometer as the indicator of land-use efficiency from the perspective of sustainability. In addition, many studies focused on the impact factors of green utilization of urban land. Li et al. [9] used the Tobit model to analyze the impact of economic development, openness, and technological progress on the green utilization efficiency of urban land. Lu et al. [10] applied the DID method to evaluate the impact of high-tech development zones on green utilization of urban land, based on the 285 cities in China from 2003 to 2016.

The LCCP policy, an important means to reduce the greenhouse effect and relieve the pressure on carbon emissions, attracted much attention during its implementation [11]. According to Li et al. [3], the LCCP policy in China specified the goals and action plans for achieving a considerable reduction in carbon emissions in the short term and for transitioning to a low-carbon economy and society in the long run. The LCCP policy not only has a profound impact on air quality [12,13,14], urban ecological efficiency [15], and the green total productivity factor [16], but is also profoundly related to the sustainability of urban land utilization. The LCCP policy will reduce carbon emissions from land utilization by optimizing the urban spatial structure [17,18]. During the implementation of the LCCP policy, the effective and appropriate matching between urban transportation network and land utilization also promotes low-carbon travel and becomes the key to the effectiveness of the LCCP policy [19,20]. There is regional heterogeneity in the influence of LCCP policy on the sustainability of urban land utilization. For example, it has been proven that there is a more obviously positive impact of LCCP on urban land utilization in cities larger in scale, with better infrastructure and better foundations for technology [16,21]. The above research provides important theoretical and methodological references to this study, but there are also some shortcomings. In terms of measurement method and index system, existing research does not fully consider the ecological factors in the process of urban land utilization. The treatment of specific indicators, such as relevant pollution emission, also needs to be improved. In addition, although the existing literature has focused on the affection of LCCP policy on the sustainability of urban land utilization, few empirical studies have been carried out on GUEUL. To bridge these gaps, this work aims to examine whether the LCCP policy results in significant positive changes in GUEUL in China. Unlike previous studies, the marginal contributions of this study are mainly in the following aspects. First, we focus on the effect of the LCCP policy from the perspective of the GUEUL, which is quite rare in the existing literature. Second, we treat the LCCP policy as a quasi-natural experiment and the propensity score matching difference-in-differences (PSM-DID) model is adopted to measure the policy’s efficacy and examine the outcomes’ robustness. It solves the endogeneity problem commonly found in the previous literature and obtains more rigorous findings.

The rest of the paper is as follows. Section 2 provides an introduction to the background of LCCP policy, methodology, and data. Section 3 addresses the results, followed by further analysis in Section 4. Section 5 is a discussion. Finally, Section 6 makes conclusions and Section 7 gives some policy implications.

## 2. Policy Background, Methodology, and Data

### 2.1. The Low-Carbon City Construction in China

Pilot has become an important testing ground for China’s low-carbon governance. The LCCP policy in China is creatively proposed as a revolutionary urban development model to resolve the conflict between economic development and environmental conservation under the background of global low-carbon transformation [16,22]. As illustrated in Figure 1, China’s LCCP policy, the key part of the country’s climate change plan, has been implemented in three batches thus far. The first batch was arranged in five provinces and eight cities by the China National Development and Reform Commission (NDRC) in 2010. During this period, the pilot areas involve multiple provincial regions and the policy content is relatively macro and not detailed enough, so that it is difficult for the LCCP policy to play an effective role. To solve this issue, the NDRC expanded the scope of low-carbon pilot areas to 28 cities and one province in 2012. The changes in pilot areas from the provincial-level city to the prefecture-level city not only considers the various regional endowments, but also makes policy implementation more realistic and relevant. At the same time, the objective of “creating a beautiful China”, stated in the report of the Communist Party of China’s 18th National Congress, has been well implemented. Continually, in order to complete the 13th five-year plan for controlling greenhouse gas emissions, the NDRC conducted the third batch in 2017 based on the favorable outcomes of the previous two batches. Without engaging the provincial-level cities, the third batch of low-carbon pilot areas covered 45 prefecture-level cities.

In the 18th National Congress of the Communist Party of China, the Chinese government committed that the carbon emissions per unit of gross domestic product will decrease by 40–45% in 2020 compared with that in 2005 [23]. To achieve this goal, the LCCP policy has four particular tasks: (1) preparing low-carbon development planning; (2) developing a low-carbon industry; (3) framing a goal assessment about reducing greenhouse gas emissions; and (4) promoting low-carbon lifestyles and green consumption patterns.

### 2.2. Model Setting

This study calculates the GUEUL based on the super-efficiency DEA model (SE-DEA). After that, the propensity score matching difference-in-differences (PSM-DID) method based on a quasi-natural experiment is adopted to identify the impact of the LCCP policy on the GUEUL.

#### 2.2.1. Super-Efficiency DEA Model (SE-DEA)

The data envelopment analysis (DEA) model is a normal and reliable method to measure GUEUL [5]. It is inevitable that in the traditional DEA model may exist multiple effective *DMUs* [24,25]. In this condition, the efficiency value is 1, which makes it impossible to access more information. To remedy this deficiency, Andersen et al. [26] developed the SE-DEA model, which allows for efficiency values greater than 1 (called super-efficiency values). The SE-DEA model can effectively distinguish and rank multiple efficient DMUs, making the GUEUL more differentiated. The model performs efficiency analysis under the condition that the production frontier of the relatively inefficient DMU (efficiency value less than 1) remains unchanged. At the same time, the model recalculates the production frontier of the DMU_S_ in the relatively efficient state and moves it backward to obtain the new efficiency value. Finally, the efficiency value calculated by the SE-DEA model is the same as that calculated by the traditional model. The linear programming equation for the SE-DEA model is as follows [27]:(1)$$\theta^{*}=MIN[\theta -\varepsilon (\sum_{i=1}^{m}S_{i}^{-}+\sum_{r=1}^{s}S_{r}^{+})]$$
(2)$$s.t.\left\{ \begin{array}{l} \sum_{\begin{array}{l} j=1\\ j\neq k \end{array}}^{n}\lambda_{j}X_{ij}+S^{-}\leq \theta X_{0},\\ i=1,2\cdots ,m\\ \sum_{\begin{array}{l} j=1\\ j\neq k \end{array}}^{n}\lambda_{j}Y_{j}-S^{+}\leq \theta X_{0},\\ r=1,2,\cdots ,s\\ \lambda_{j}\geq 0,j=1,2,\cdots ,n;S^{-}\geq 0;S^{+}\geq 0 \end{array}\right.$$

Assume that there are *n DMUs* to be evaluated and the known crisp values of inputs X and outputs Y are non-negative; θ is the parameter to be determined; λ is an intensity vector of *DMUs*; $S_{i}{}^{-}$ and $S_{r}{}^{+}$ are the slack variables, respectively. $S^{-}$ indicates the likely reduced amount of input and $S^{+}$ denotes the possibly increased amount of input, used to transform the inequalities equations into equivalent ones. The solution to the model is denoted by $\theta^{*}$. The state of *DMUs* can be determined based on the following three situations: (1) If $\theta^{*}=\theta \neq 1$, the evaluated *DMU* is inefficient. (2) If $\theta^{*}=\theta =1$ and all slack variables $S_{i}^{-}=S_{r}^{+}=0$, the evaluated *DMU* is strongly efficient. (3) If $\theta^{*}=\theta =1$ and all slack variables $S_{i}^{-}=0$ or $S_{r}^{+}=0$ for at least one i and r, the evaluated *DMU* is weakly efficient. 

#### 2.2.2. Propensity Score-Matching Method (PSM)

It is assumed that the LCCP policy is a quasi-natural experiment. The difference-in- differences (DID) model can be employed to study the effect of policy on the GUEUL [28,29]. The DID model requires a completely random selection between treatment group and control group; otherwise, the result can be largely biased. In this study, cities that implement the LCCP policy are the treatment group, while the others are the control group. However, the selection of pilot cities is not actually random, which is based on many factors, such as the geographical location, economic development, population density, and environmental constraints [30]. Therefore, it is necessary to eliminate the bias caused by the self-selection problem with the help of the PSM model before using the DID model.

The PSM model is a type of statistical method that uses non-experimental data or observational data for interventional effect analysis [31]. It can shift from matching based on treatment covariates $X_{i}$ to matching based on one-dimensional propensity index $X=\left(X_{1},X_{2},\cdots ,X_{n}\right)$. In addition, the probability that an individual with the characteristics D will receive the treatment can be predicted. The formula for estimating the propensity index is as follows:(3)$$p\left(X_{i}\right)=E\left(D_{i}/X_{i}\right)=Pr\left[D_{i}=1/X_{i}\right]$$
where $p\left(X_{i}\right)$ is the probability that an individual with characteristic $X_{i}$ receives the treatment $\left(D_{i}=1\right)$. The samples after matching need to meet the common support assumption test and the balance test. Firstly, we test the common support assumption, which is that individuals with the characteristic $X_{i}$ will receive treatment, between 0 and 1. The expression is as follows:(4)$$0<\mathrm{Pr}\left[D_{i}=1/X_{i}\right]<1$$

Secondly, the balance test is to examine whether there is a significant difference in the covariate values between treatment group and control group after matching. If the difference is not significant, the matching effect is good and it is more appropriate to use such matched samples for further analysis. The bias is calculated by the following formula:(5)$${\hat{△}}_{ct}\equiv \frac{\left|{\bar{X}}_{t}-{\bar{X}}_{c}\right|}{\sqrt{\left(s_{t}^{2}+s_{c}^{2}\right)/2}}$$
where ${\bar{X}}_{t}$ is the mean value of a covariate in the treatment group; ${\bar{X}}_{c}$ is the mean value of a covariate in the control group; $s_{t}^{2}$ is the sample variance of a covariate in the treatment group; and $s_{c}^{2}$ is the sample variance of a covariate in the control group. If ${\hat{△}}_{ct}$ is close to 0, it means that the distribution of the two groups is more balanced.

#### 2.2.3. Difference-in-Differences Model (DID)

The DID model will use the new treatment and control groups the PSM model generated. It investigates the difference in GUEUL between the treatment group and the control group before and after the implementation of the LCCP policy, that is, the net effect on the GUEUL brought by the LCCP policy. The estimation function is set as follows:(6)$$GUEUL_{i,t}=\alpha +\beta LCCP_{i,t}+\gamma CV_{i,t}+\mu_{i}+v_{t}+\varepsilon_{i,t}$$
where GUEUL, the green utilization efficiency of urban land, is the dependent variable. i and t represent prefecture-level cities and years, respectively. $LCCP_{i,t}$ is the policy dummy variable, set at “1” if the city is approved as pilot city in or after the year *t* and “0” otherwise. Additionally, we include a series of relevant control variables, represented by $CV_{i,t}$, indicating other possible factors influencing *GUEUL*. α is a constant term. β is the core parameter to be estimated, representing the effect of the LCCP policy on the GUEUL; γ is the coefficient of the control variables; $\mu_{i}$ is the city fixed effect, representing an effect that changes with city but not with time; $v_{t}$ is the time fixed effect, indicating an effect that changes with time but not with city; and $\varepsilon_{i,t}$ is the error term.

### 2.3. Variable Selection and Description

The main focus in this study is the effects of LCCP on GUEUL. Additionally, many factors also affect GUEUL, so some control variables are added. The concrete methods for variable selection and processing are presented in Table 1.

#### 2.3.1. Dependent Variable

The dependent variable in this study is GUEUL, measured by the SE-DEA model. Urban land-use efficiency is measured using the conventional DEA method, which excludes ecological considerations. The land-use process has other inputs, in addition to economic ones. We must also include the emissions of various pollutants as a result of land usage when calculating efficiency. The GUEUL is a comprehensive reflection of the integrated inputs and outputs of the urban land-use system under certain production technology conditions [32]. Its goal is to maximize the green economic outputs in the land at the cost of the least possible inputs and ecological losses. Accordingly, based on the traditional measurement method of land utilization efficiency, this study considers the ecological inputs in land utilization. In particular, the pollution emission index is incorporated into the measurement model of GUEUL. GUEUL is better able to gauge how well excavated land is being used. The assessment of GUEUL in contemporary cities has steadily developed into a research hotspot in the present land-use evaluation due to the maturing of the idea of green development and the advancement in research methodologies and technology.

Guided by the philosophy that lucid waters and lush mountains are invaluable assets, we construct a comprehensive evaluation index system for GUEUL in China, which includes input variables and output variables. Input variables include: (1) Land input: the proportion of built-up area in urban area [33]. Land is a crucial component in production processes and the fundamental building block in urban development. The proportion of built-up area in urban area is a crucial metric for assessing the rate of urban growth, the level of development, and GUEUL. (2) Workforce input: the number of employees in the tertiary industries [34]. In comparison to other industries, the tertiary sector has developed green land use more successfully. The number of employees in the tertiary industries has a significant influence on GUEUL and urban construction. (3) Capital input: the amount of new fixed assets in urban municipal public facilities [35]. Capital input affects the method of land use. The amount of new fixed assets of urban municipal public facilities is a symbol of the local government’s ability to provide public goods and services. Thus, capital input has an impact on the optimal allocation policy of land resources and the intensification of land use. (4) Ecological input: Three major sources of pollution in cities are selected as the original indicators of ecological input: the urban pollution from the industrial wastewater discharge, industrial SO_2_ emissions, and industrial soot emissions [36]. In order to eliminate the difference of magnitude caused by different measurement units of three kinds of ecological input, the comprehensive index of ecological input is calculated by using the entropy weight method [37]. (5) Economic output: the added value of the secondary and tertiary industries. In order to reflect the actual situation of social economy, the added value of the secondary and tertiary industries is used as the evaluation index for economic output [38].

#### 2.3.2. Independent Variable

As shown in Equation (6), in this study, the policy dummy variable $LCCP_{i,t}$ is the independent variable. If the value is 1, it means that the sample is a pilot city; otherwise, it is 0. LCCP is currently implemented in three batches. The first, in 2010, covered 5 provinces and 8 cities, for a total of 82 cities. The second, in 2012, covered 1 province and 28 prefecture-level cities, and the third batch, in 2017, covered 45 prefecture-level cities. In addition, it is worth noting that there is an overlap in the pilot lists; that is, Wuhan, Guangzhou, Kunming, and Yan’an appear in both the first and second batch. Studies have pointed out that if a province implements LCCP policy, then the cities under its jurisdiction are deemed to participate in this policy and the implementation time is set to an earlier one [16].

#### 2.3.3. Control Variables

The selection of the control variables is mainly based on the literature on the driving factors for GUEUL [25,39,40]. Specifically, it includes: (1) Urbanization (Urban), measured by the proportion of the non-agricultural population to the total population [41], which affects the rate of spatial expansion and utilization pattern of towns and cities, thus, changing the urban land-use structure. (2) Degree of openness (FDI), calculated by total foreign direct investment as a percentage of GDP [42], which affects the degree of urban economic development and enables the urban land system to gain development momentum from outside. (3) Industrial structure (Ind2), measured by the percentage of secondary industry [43]. The adjustment in industrial structure drives the transformation of land-use structure and directly affects the development direction of urban land resource allocation. (4) Population density (Pop), the ratio of the total population to the administrative area [44], which is an important indicator to reflect whether the structure of urban land-use is consistent with the size of the population and if the area of land for construction per capita is too large, the overall land use in the city is rough. (5) Urban Ecology (UE), calculated by the green space per capita [45], which can reflect the environmental quality of residents’ life and the degree of protection of urban ecological environment. (6) Land resource conditions (Land), measured by the construction land area per capita [46]. An increase in per capita land area can lead to economic growth, but it also increases waste emissions and the overexploitation of natural resources. (7) Government support (Gov), measured by the budgeted government expenditure as a percentage of GDP [47], which can have a profound impact on land elements.

#### 2.3.4. Data Description

Due to data availability, we selected 284 prefecture-level cities in China as the sample. The study period is from 2006 to 2019. These data are from China City Statistical Yearbook (CCSY), China Urban Construction Statistical Yearbook (CUCSY), the Provincial Statistical Yearbooks (PSY), and the government websites for each city. For missing data, we use the mean interpolation method to provide estimated values. Further, we take the natural logarithm of all continuous variables, except for the percentage values, which can weaken heteroscedasticity and avoid the effect of outliers [48]. Table 2 shows the descriptive statistics for each variable.

## 3. Results and Analysis

### 3.1. Application of the PSM Method

Through the SE-DEA model, we calculate the value of GUEUL, and the average value of GUEUL from 2006 to 2019 is not high. This value is in a continuous increasing trend from 0.062 to 0.155, with the most significant increase in the value from 2015 to 2016. The overall growth rate of GUEUL increases from 4.26% to 10.84%, but the growth rate is negative in 2009, 2011, 2014, and 2017. Figure 2 shows the trend of GUEUL for low-carbon pilot cities (treatment group) and non-pilot cities (control group) from 2006 to 2019. In terms of time, since 2006, GUEUL has shown an upward trend as a whole, whether it is a low-carbon pilot city or a non-pilot city. What is striking in this figure is that the GUEUL of the pilot cities is always greater than that of the non-pilot cities, but the trend of change appears different. Specifically, the time points of the policy implementation are 2010, 2012, and 2017, and we mainly observe the first policy shock point (in 2010). The figure shows that the GUEUL of pilot cities and non-pilot cities has opposite development trends before the policy implementation and the trend becomes the same after the policy implementation. It could be argued that the changes in GUEUL were brought about by the LCCP policy. Before 2010, the GUEUL of the pilot cities showed a downward trend and the GUEUL of the non-pilot cities showed an upward trend. After 2010, the GUEUL of pilot cities became an upward trend and the non-pilot cities still maintained an upward trend. The following will take a rigorous identification from the empirical data to verify.

We first conduct a regression analysis based on the PSM method. Figure 3 shows the results of the common support assumption test, which is the distribution of propensity scores in the treatment and control groups. The results suggest that 38 cities in the control group failed to meet the common support assumption. These cities either have an extraordinarily higher or lower probability of being selected as pilot cities. Therefore, we exclude them from the samples.

The balance test is displayed in Table 3 and Figure 4, which reflects the propensity scores and degree of deviation in the variables, respectively. According to Rosenbaum and Rubin [49], if the absolute value of the normalized bias value (|Bias| (%)) (abbreviation: the bias) of the matched variable is significantly less than 20%, the matching estimation results are considered to be reliable. In Table 3, we can see that the bias value of all the matched variables is almost less than 10%. At the same time, the t-test after matching is not significant. This indicates that there is no significant difference between the matched treatment and control group and avoids sample selection bias. The results in Figure 4 are consistent with those in Table 3; the bias values are all less than 10% and significantly smaller than those of the unmatched values.

Further, Figure 5a,b reflect the kernel density of the propensity score (P-score) values before and after matching, respectively. As can be seen from Figure 5a, before matching, the probability density distributions of the P-core have great difference between the two groups. After matching, however, the consistency of the probability density distribution increased (see Figure 5b). This indicates that the sample similarity of the two groups improved after matching. The matching results can be safely used for further DID estimation.

### 3.2. Main Regression Results

In order to ensure the robustness of the results, we first show the basic regression results of the DID models with unmatched data in columns (1) and (2) in Table 4, without and with control variables, respectively. The independent variables all satisfy 95% confidence intervals, indicating that the LCCP policy significantly promoted the GUEUL. Then, columns (3) and (4) report the regression results without and with control variables based on the PSM matched data, respectively. The results show that compared with the non-pilot cities, the GUEUL of pilot cities is about 2.66% higher after controlling for other variables. The regression results in both the DID model and the PSM-DID model robustly show that the LCCP policy obviously promoted the GUEUL. The finding is consistent with the existing literature, which affirms the positive effects of the LCCP policy. For example, Wolff [50] identified that the low-carbon-zone policies significantly improved local air quality. Qiu [51] also found the LCCP policy had, indeed, exerted a positive effect on the green total factor productivity in pilot cities and the effect would increase over time.

As for control variables in column (4), we can see that urbanization, degree of openness, industrial structure, land resource conditions, and government support are important factors in influencing GUEUL. Specifically, the impact of urbanization on GUEUL is not statistically meaningful. The degree of openness has a depressive effect on GUEUL, which is consistent with Zhao et al. [52]. They found that China had not opened its market adequately and foreign investment agglomeration played only a limited role in improving urban land-use efficiency. The smaller the proportion of total output value of the secondary industry in a city, the more able to improve GUEUL. The likely reason for that is that the secondary industry will bring high pollution and high consumption. The improvement in urban ecology fails to significantly contribute to GUEUL. As the construction land area per capita grows, GUEUL declines at a rate of 0.541, at a significance level of 1%, which may require the government to reconsider land-use planning and tap internal land stocks from a domestic perspective [53,54]. In terms of government support, one unit increase in the government financial expenditure as a percentage of GDP will decrease the GUEUL of the pilot city by 5151. Government spending on green technologies is not entirely effective [55].

### 3.3. Dynamic Effects Test

To test the dynamic effects, this study draws on the literature of Beck et al. [56], Li et al. [57], and Gehrsitz and Markus [58] to conduct dynamic effects tests by Event Study Approach (ESA) and construct the following econometric model:(7)$$\begin{array}{cc} GUEUL_{i,t} & =\alpha +\delta_{1}LCCP_{i,t}^{-4}+\delta_{2}LCCP_{i,t}^{-3}+\cdots +\delta_{9}LCCP_{i,t}^{+4}+\delta_{10}LCCP_{i,t}^{+5}\\ & +\gamma CV_{i,t}+\mu_{i}+v_{t}+\varepsilon_{i,t} \end{array}$$
where $LCCP_{i,t}^{-j}$ and $LCCP_{i,t}^{+j}$ represent dummy variables in the *j*-th years before and after the implementation of the policy, respectively. δ is the parameter to be estimated, representing the effect of LCCP on the GUEUL. The meaning of the remaining variables is consistent with Equation (6). If the regression coefficients of $\delta_{1}∼\delta_{10}$ are not significant, it means that there is no significant difference between the treatment and control groups before the policy shock and the sample satisfies the parallel trend test. The paper studies data from 2006 to 2019, taking 2010 as the occurrence point of the policy, covering 4 years before the implementation and 10 years after the implementation. For easy observation, this paper is set as the first 4 and the last 5. We exclude the first year before implementation, estimating the dynamic changes in the differences of LCCP on GUEUL between the treatment and control groups in the study period.

In order to visually test the hypothesis of common trend and observe the dynamic impact of low-carbon city construction on GUEUL effect, Figure 6 is drawn to show the difference in GUEUL before and after the LCCP policy [59,60]. This graph illustrates that none of the regression coefficients were significant before the policy implemented, which indicates that there is no difference in GUEUL between the treatment group and the control group before the LCCP policy. There is no significant difference between the treatment and control groups before policy implementation; thus, the parallel trend hypothesis holds. In the fourth year and after the implementation of the policy, the regression parameters are all positive and significant, indicating that the impact of the LCCP policy on GUEUL has a certain time lag.

### 3.4. Placebo Test

This study uses an indirect placebo test to process the effect of unobservable characteristics in the PSM-DID model [61,62]. Specifically, a certain number of samples is randomly selected as treatment group, referred to as pseudo-policy dummy variables. We first control a set of key and observable urban characteristics, including urbanization, degree of openness, industrial structure, land resource conditions, urban ecology, population density, and government support. Secondly, we observe whether the kernel density of the independent variable’s coefficient and *p*-values are concentrated around 0 and significantly deviate from their true values. The distributions of coefficient and *p*-value are shown in Figure 7 and they are distributed around 0, indicating that most of the coefficients of the “pseudo-policy dummy variable” are not significant, so the results of the benchmark regression are robust.

## 4. Further Analyses

There are huge differences in geographic location, economic level, and resource endowment between cities. These differences may lead to different responses to LCCP policy in different cities. On the one hand, the unbalanced development in China’s regional economies can lead to different effects of the implementation of pilot low-carbon city policies in different regional cities. On the other hand, different resource-based city types have different energy consumption and industrial structures and their low-carbon development is a coordinated process of economic, social, and ecological systems. To explore more information about the influence of these factors on the regression results, we classified the samples according to the regional socioeconomic [63] and resource endowment level [64].

### 4.1. Regional Economic Heterogeneity

According to the classification criteria of the 5th Session of the 8th National People’s Congress in 1997, we divided the 284 cities into four groups, namely, the eastern, central, western, and northeastern area and set the following model:(8)$$GUEUL_{i,t}=\alpha +\lambda LCCP_{i,t}\times LOCA_{i,t}+\gamma CV_{i,t}+\mu_{i}+v_{t}+\varepsilon_{i,t}$$
where $LOCA_{i,t}$ is a dummy variable, which represents the regional economics by referring to Attavanich et al. [65]. The coefficient of interaction term λ can capture the effect of the LCCP policy on GUEUL in different regions. If the city is in the east, the value of regional dummy variable is “1”; otherwise, it is “0” [66]. The same rule is applied to the sample in the western, central, and northeastern area.

Table 5 provides the heterogeneity test results for regional economics. From columns (1), (2), (5), and (6), we can see a significant and positive correlation exists in the eastern and western areas. What is striking in this table is that the coefficient value of the LCCP in the eastern area is larger than that in the western area. As demonstrated in columns (3) and (4), in the central area, there is a significant inhibitory effect of the LCCP policy on GUEUL without control variables; however, the addition of control variables has no significant effect on GUEUL. In the northeastern area, LCCP reflects a significant inhibitory effect on GUEUL, whether or not control variables are considered.

### 4.2. Resource Endowment Heterogeneity

Urban resource endowment plays an important role in policy implementation. This study sets dummy variables to signify resource-based cities. According to the National Plan for Sustainable Development of Resource-Based Cities (2013–2020), we further divide the 284 cities into four types: growing (*Resg*), maturing (*Resm*), declining (*Resd*), and regenerating (*Resr*). We introduce the variable and set the following model:(9)$$GUEUL_{i,t}=\alpha +\lambda LCCP_{i,t}\times Res_{i,t}+\gamma CV_{i,t}+\mu_{i}+v_{t}+\varepsilon_{i,t}$$
where Res refers to the dummy variable, namely, resource endowment; the coefficient λ is the estimated parameter. The regression results in Table 6 show that the LCCP policy benefits GUEUL in growing resource-based pilot cities. Meanwhile, the same rule is applied to the maturing and regenerating resource-based pilot cities, showing that the LCCP has little influence on GUEUL. In addition, in areas of declining resource-based pilot cities, there is a negative influence in GUEUL by LCCP at a significance level of 10%.

## 5. Discussion

In this study, we selected 284 cities in China as a sample to explore the impact of LCCP on GUEUL based on the PSM-DID model.

This study found that the LCCP policy has a positive effect on GUEUL in China, which is consistent with the results of Song et al. [25]. They suggest that the economic development in the pilot cities had not been at the expense of the ecological environment. The LCCP policy has not only a direct but also an indirect impact on GUEUL. The construction of low-carbon cities leads to changes in growth targets, which allows the cities to gradually transform towards quality-centered inclusive development and then increases the level of GUEUL [67]. Our findings are inconsistent with Greenstone et al. [66] and Liu et al. [68], who found that strict LCCP policies increase production costs and innovation inputs within a short time through the compliance costs effect and crowding effect, then causing a decline in the GUEUL. Over a long time scale, the LCCP policy may affect GUEUL through three intermediate effects, which are land-use planning, industry structure, and technological innovation. First, the mandate of the LCCP policy clearly states that local governments need to integrate the concept of low-carbon development into land-use planning, which allows the urban land to play a better role in coordinating economic development and ecological protection. Second, the tertiary, low-carbon and environmental-protection industries get a rapid development after the construction of low-carbon cities [69], which is conductive to promote the specialization in the division of industrial labor, the scale of production, and the coordinated development of industries, thereby raising the realization of scale economics and reducing pollution and energy consumption [70]. Last but not least, in research and development (R&D) and emerging fields, with the advancement in the LCCP policy, enterprises realize that environmental regulations are long term and inevitable [16]. Thus, it is more efficient to invest in emission reduction technologies in advance [71,72].

We found that the LCCP policy has a significant positive impact on GUEUL in eastern and western areas, while pilot cities in the central and northeastern areas showed inhibiting effects. The effects of the LCCP policy rely on crucial elements of local economy, governance, and technology. With the introduction of the LCCP policy, the incentive for cities to minimize the growing cost of pollution through technological innovation increases [73]. Pilot cities in the eastern area have a higher level of economic output, human capital, and technological foundation, making low-carbon technology innovation more accessible [74]. The construction of low-carbon cities encourages the growth of green enterprises, as well as the protection and preservation of natural resources, especially land resources. Pilot cities in the western area with plentiful renewable sources and extensive land may modify their energy structure [75], resulting in more efficient land-use planning and a higher level of GUEUL. However, the central region faces many problems in constructing low-carbon cities, such as low awareness of low-carbon life among citizens, difficulties in adjusting high-energy-consuming industries, and environmental risks in undertaking industrial transfer [76]. These problems also hinder the increase in GUEUL as well. In the future, the central region needs to take advantage of its rich resources and location to innovate low-carbon development of the urban land, to get rid of the “central depression” and realize the “central rise”. The main difficulties in constructing low-carbon cities in the northeast region are resource exhaustion, single industrial structure, and difficulties in low-carbon transformation [77]. Particularly, the northeast has more densely populated cities, more outdated industrial facilities, aged business equipment, and lagging technological advancement. The region as a whole lacks the internal strength to turn GUEUL into a low-carbon metropolis and the necessary resources and technologies [51].

There was a significant positive relationship between the LCCP policy and GUEUL in growing resource-based pilot cities, while having a negative relationship in declining resource-based pilot cities and no obvious relationship in mature and regenerative resource-based pilot cities. Growing resource-based pilot cities are still in the early stages of resource development [78], whose potential for resource security and socio-economic growth is much higher than other areas [79]. A specific task for the LCCP policy is to rationalize resource planning with a low-carbon development concept and then promote firms in the region to conserve energy and decrease emissions. As for declining resource-based pilot cities, they are defined by depleted resources, delayed economic development, and enormous ecological and environmental issues [80]. In such cities, incentives for industrial transformation and smart land allocation are weak. Pilot arrangement is unlikely to help such cities improve their situation in the near future. Instead, it will raise the cost of pollution reduction. With the continuous exploitation of resources, maturing resource-based pilot cities are in a strong position in the national energy resource supply. At the same time, the regenerative resource-based pilot cities are virtually resource-free and expand in a virtuous manner, so the same environmental restrictions may no longer be applicable [81].

Although this study presented initial evidence on the positive effect of the LCCP policy on GUEUL, several deficiencies that can be addressed in future studies need to be acknowledged. Due to data availability, when calculating the level of GUEUL, some meaningful variables are excluded from input and output variables, such as the innovative technology potential, the social welfare improvement, and the ecological benefits. Although we addressed the intermediary effects of the LCCP policy on GUEUL, additional in-depth case studies are required to fully understand the shaping of the intermediary effects. Although the PSM-DID model can provide a fairly precise way to estimate the overall impact of LCCP, it can still be improved in terms of capturing policy spillover effects. In future studies, we need to calculate GUEUL based on more inclusive indexes, which could make the results more precise, and investigate the intermediary effects of the LCCP policy more thoroughly, adding a spatial regression analysis model to fill the gap.

## 6. Conclusions

The pursuit of GUEUL has become increasingly prominent in government agendas [81]. Based on statistical analysis, we regard adopting and implementing the LCCP policy as an exogenous policy shock and introduced the PSM-DID method to estimate the policy effects. In addition, we conducted further analyses on the heterogeneous influences from the perspective of regional economic and resource endowment. The main conclusions drawn from the empirical analyses are as follows. First, the implementation of the LCCP policy significantly improves GUEUL and can effectively increase the possibility of green and low-carbon development in cities in the future. Second, the effects of the LCCP policy on GUEUL are varied due to differences in regional economic and resource endowment. In terms of regional economics, the implementation of the LCCP policy in the eastern and western areas leads to a significant improvement effect on GUEUL, while the central and northeast areas face opposite effects. In terms of resource endowment, compared with the maturing, declining, and regenerating resourced-based pilot cities, the LCCP policy can better improve GUEUL in growing resourced-based pilot cities.

## 7. Implications

The results of our research could provide direction and evidence for the optimization of the LCCP policy and provide reference for other cities around the world to promote green and low-carbon development, especially about GUEUL. Based on the study findings, the following policy suggestions are offered.

First, governments should regard the construction of low-carbon cities as a long-term policy orientation rather than short-term management [82], given the time lag effect of LCCP policy on GUEUL. This is critical to the quality of low-carbon city construction and the degree of collaboration between the central government and local governments [80]. On the one hand, the central government, as the leader and regulator in the LCCP policy, should define the pilot city selection, allowing for a gradual increase in the number of demonstration cities [83]. Financial backing, talent management, and other incentives from the central government to pilot regions would motivate and guide local governments [84]. More significantly, the central government should create a scientific evaluation index from the perspective of GUEUL, allowing the local government to implement the policy in a targeted manner [85]. On the other hand, local governments, the major participants in the implementation of the LCCP policy, should actively cooperate with the decision-making from the central government to take various measures to ensure the construction of low-carbon cities [86,87]. During the construction of low-carbon city, the improvement in GUEUL relies substantially on scientific land-use and transport planning, industrial upgrades, green technology innovation, and talented persons [88,89,90]. Local governments must perform scientific and preferential policies to promote the smooth progress of low-carbon development [91].

Second, since the performance of the LCCP policy is linked to city features, local governments should adjust its low-carbon development strategy scientifically according to those differences. Firstly, central government should consider differences in critical socioeconomic factors to select the pilot cities. Furthermore, the central and northeastern cities can be allocated more financial security and new energy development opportunities [92]. Due to the diversity of regional economics, the areas that experience significant GUEUL gains after policy implementation are more likely to be economically developed areas or have plentiful renewable resources and extensive land areas. In addition, government is necessary to encourage the industrial structure to transform and upgrade in the direction of green and low-carbon growth through technological innovation, economical and intensive land-use planning, low-carbon industry and lifestyle development, and so on [93,94]. The development of resource-based cities is closely related to changes in industrial structure, ultimately reversing resource-based cities’ unreasonable dependence on and unsustainable use of land resources.

## Figures and Tables

**Figure 1 ijerph-19-09844-f001:**
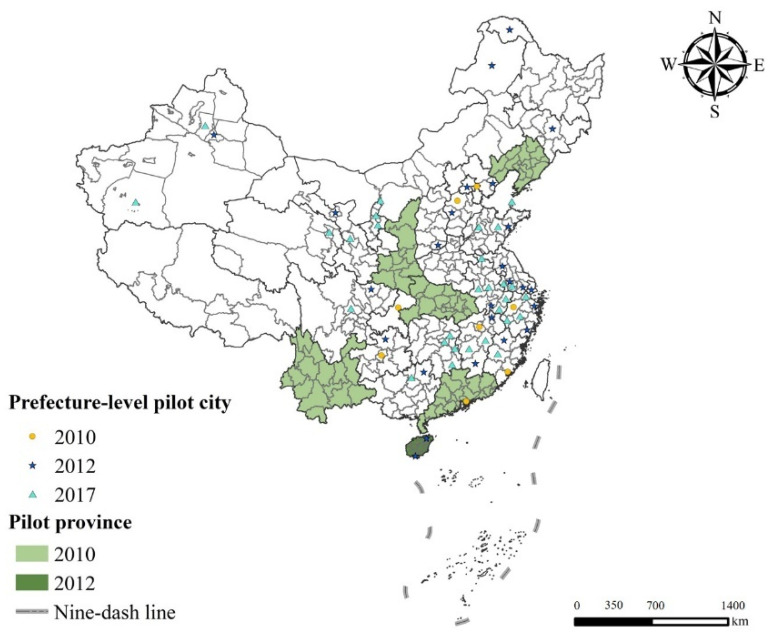
Spatial distribution of three batches of low-carbon pilot areas in China.

**Figure 2 ijerph-19-09844-f002:**
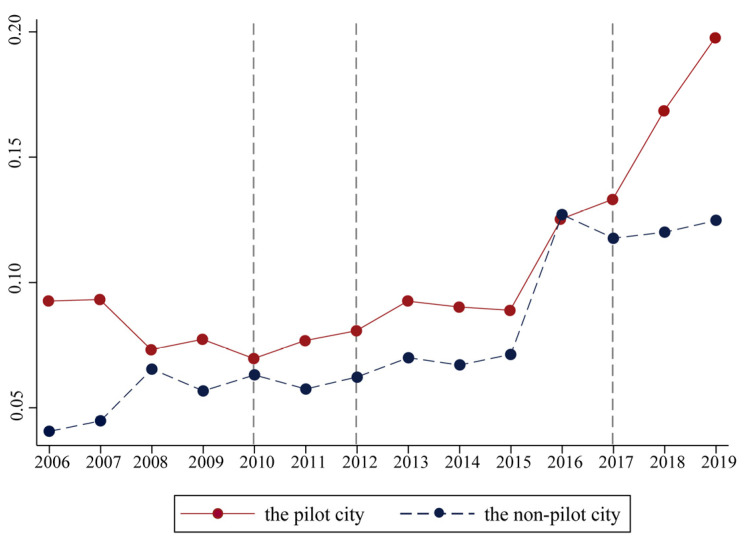
The GUEUL in the pilot city and non-pilot city.

**Figure 3 ijerph-19-09844-f003:**
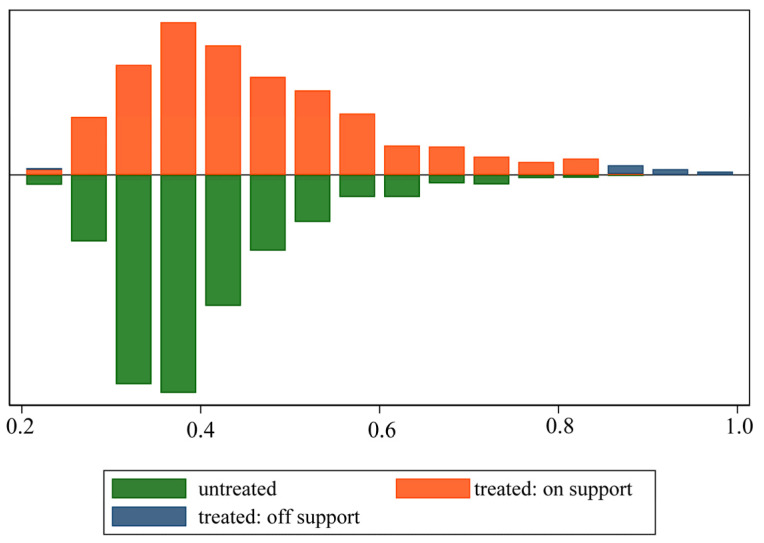
The distribution of propensity scores for the treatment and control groups.

**Figure 4 ijerph-19-09844-f004:**
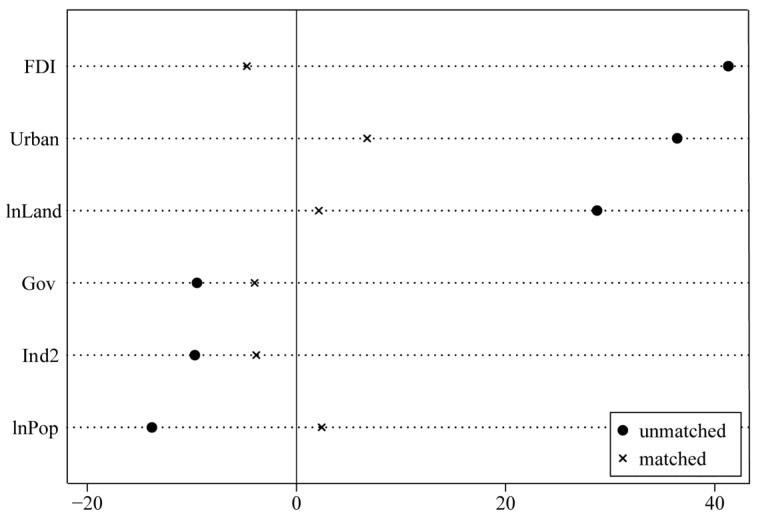
The deviation in each covariate for the treatment and control groups.

**Figure 5 ijerph-19-09844-f005:**
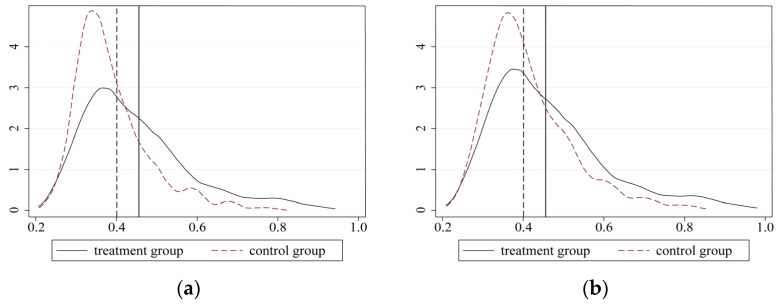
The P-score kernel density. (**a**) P-score before matching. (**b**) P-score after matching.

**Figure 6 ijerph-19-09844-f006:**
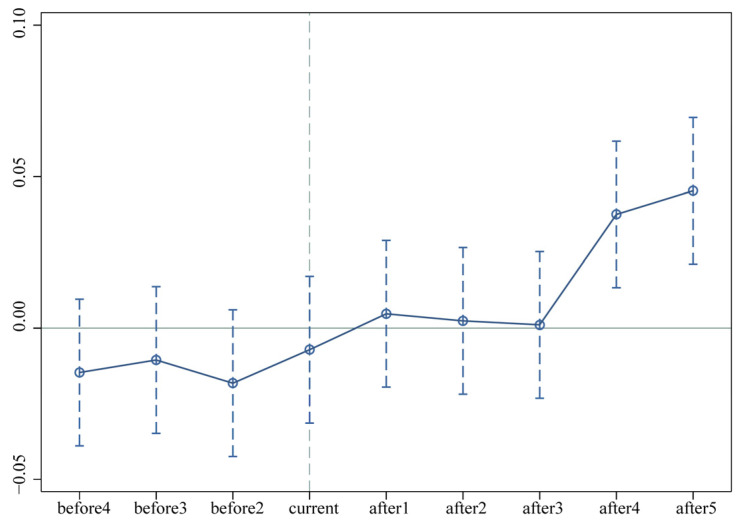
The dynamic effect tests.

**Figure 7 ijerph-19-09844-f007:**
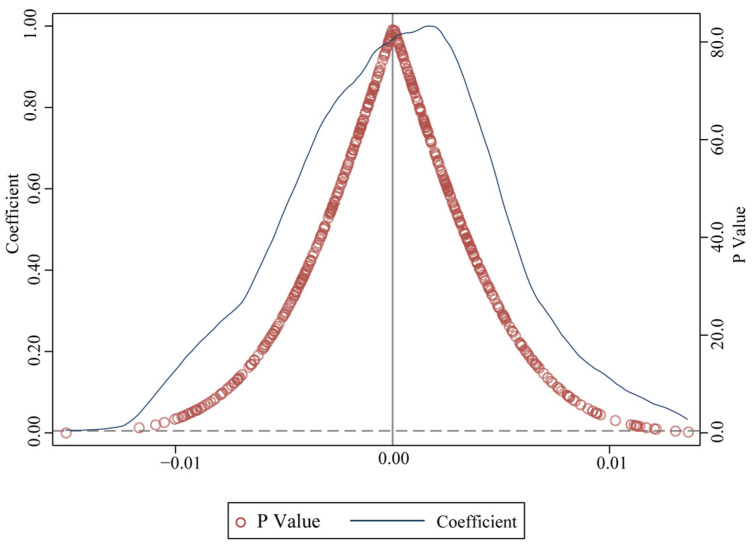
Kernel density of placebo test.

**Table 1 ijerph-19-09844-t001:** Primary variables and processing methods.

Variable Type	Symbol	Variable Name	Processing Methods
Dependent variable	GUEUL	Green utilization efficiency of urban land	Super-efficiency DEA (SE-DEA) model
Independent variable	LCCP	Low-carbon city pilot	Dummy variable
Control variables	Urban	Urbanization	(The population of the non-agricultural/total population) × 100%
	FDI	Degree of openness	(The actual FDI in the region/regional GDP) × 100%
	Ind2	Industrial structure	(The actual total foreign direct investment in the region/regional GDP) × 100%
	Pop	Population density	Regional total population at the end of the year/administrative area
	Env	Urban Ecology	Regional green space area/total population in region
	Land	Land resource conditions	The regional construction land area/total population in region
	Gov	Government support	(Budgeted government expenditures/regional GDP) × 100%

**Table 2 ijerph-19-09844-t002:** Descriptive statistics.

Variable Type	Symbol	Sample Size	Mean	Standard Deviation	Min.	Max.
Dependent variable	GUEUL	3990	0.09	0.13	0.00	2.93
Independent variable	LCCP	3990	0.43	0.50	0.00	1.00
Control variables	Urban	3990	46.54	18.39	4.43	100.00
	FDI	3990	20.22	32.66	0.00	324.20
	Ind2	3990	47.22	11.11	10.03	99.15
	lnPop	3990	7.92	0.84	3.66	9.91
	lnUE	3990	14.08	0.84	9.40	17.85
	lnLand	3990	15.55	1.52	11.05	22.31
	Gov	3990	18.32	10.61	0.60	95.19

**Table 3 ijerph-19-09844-t003:** Comparisons of sample characteristics of unmatched and matched in PSM.

Variable	Unmatched/	Mean		Bias (%)	Reduction of Bias (%)	*t*-Test	
	Matched	Treatment	Control			t	*p* > *t*
Urban	U	50.37	43.67	36.40		11.59	0.00
	M	49.66	48.41	6.80	81.40	1.95	0.05
FDI	U	28.10	14.32	41.30		13.48	0.00
	M	24.20	25.78	−4.70	88.50	−1.37	0.17
Ind2	U	46.61	47.67	−9.70		−3.00	0.00
	M	46.54	46.96	−3.80	60.50	−1.08	0.28
lnPop	U	7.85	7.96	−13.80		−4.32	0.00
	M	7.85	7.83	2.40	82.50	0.68	0.50
lnLand	U	15.79	15.36	28.70		8.98	0.00
	M	15.75	15.72	2.10	92.50	0.62	0.54
Gov	U	17.74	18.75	−9.50		−2.98	0.00
	M	17.84	18.27	−4.00	58.00	−1.16	0.25

**Table 4 ijerph-19-09844-t004:** The impact of LCCP city on the GUEUL.

Variable	DID	DID	PSM-DID	PSM-DID
	(1)	(2)	(3)	(4)
LCCP	0.0380 ***	0.0258 ***	0.0350 ***	0.0264 ***
	(7.78)	(5.21)	(7.88)	(6.01)
Urban		0.0282		0.0175
		(1.83)		(1.22)
FDI		−0.429 **		−0.474 ***
		(−2.85)		(−3.33)
Ind2		−7.541 ***		−6.457 ***
		(−9.06)		(−8.34)
lnUE		0.624 *		0.484
		(2.30)		(1.89)
lnLand		−0.406 *		−0.541 **
		(−2.19)		(−3.12)
Gov		−5.064 ***		−5.151 ***
		(−10.84)		(−11.70)
Cons	0.0791 ***	0.638 ***	0.0744 ***	0.647 ***
	(32.62)	(9.86)	(32.38)	(10.43)
*N*	3990	3990	2884	2884

Note: *t* statistics in parentheses; *, ** and *** represent *p* < 0.05, *p* < 0.01, and *p* < 0.001, respectively.

**Table 5 ijerph-19-09844-t005:** Regional economic heterogeneity test results.

Variable	LCCP × Eastern		LCCP × Central		LCCP × Western		LCCP × Northeastern	
	(1)	(2)	(3)	(4)	(5)	(6)	(7)	(8)
LCCP	0.060 *** (10.87)	0.058 *** (9.15)	−0.016 * (−2.43)	−0.012 (−1.71)	0.015 * (2.18)	0.025 *** (3.55)	−0.031 *** (−3.36)	−0.026 ** (−2.79)
Control variables	NO	YES	NO	YES	NO	YES	NO	YES
Year fixed effect	YES	YES	YES	YES	YES	YES	YES	YES
Constant	0.079 *** (35.32)	0.465 *** (6.88)	0.090 *** (41.34)	0.519 *** (7.62)	0.087 *** (39.84)	0.535 *** (7.86)	0.090 *** (42.53)	0.513 *** (7.54)
N	3990	3990	3990	3990	3990	3990	3990	3990

Note: *t* statistics in parentheses; *, ** and *** represent *p* < 0.05, *p* < 0.01, and *p* < 0.001, respectively.

**Table 6 ijerph-19-09844-t006:** Resource endowment heterogeneity test results.

Variable	LCCP × Resg		LCCP × Resm		LCCP × Resd		LCCP × Resr	
	(1)	(2)	(3)	(4)	(5)	(6)	(7)	(8)
LCCP	0.048 ** (3.08)	0.050 ** (3.09)	−0.006 (−0.73)	−0.002 (−0.20)	−0.043 *** (−3.42)	−0.031 * (−2.44)	−0.000 (−0.00)	0.011 (0.61)
Control variables	NO	YES	NO	YES	NO	YES	NO	YES
Year fixed effect	YES	YES	YES	YES	YES	YES	YES	YES
Constant	0.088 *** (42.11)	0.511 *** (7.50)	0.089 *** (41.58)	0.522 *** (7.67)	0.090 *** (42.87)	0.507 *** (7.43)	0.089 *** (42.54)	0.525 *** (7.69)
N	3990	3990	3990	3990	3990	3990	3990	3990

Note: *t* statistics in parentheses; *, ** and *** represent *p* < 0.05, *p* < 0.01, and *p* < 0.001, respectively.

## Data Availability

The data will be made available to the reader upon request.

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
