# Peer review of "Has China’s Low-Carbon City Construction Enhanced the Green Utilization Efficiency of Urban Land?"

_ijerph, 2022, doi:10.3390/ijerph19169844_

Round 1

Reviewer 1 Report

In this article, the authors focus on an interesting topic: the impact of the low-carbon city pilot (LCCP) policy on the green utilization efficiency of urban land (GUEUL) in China. The structure is well organized and the paper is well written, However, I think that could include some improvements before being accepted for publication.

1) The relationship between LCCP policy and GUEUL should be theoretically clarified in the introduction section, in particular, the authors should distinguish the two terms “low carbon” and “green”, commonly used in the field of urban land use research, by explaining why GUEUL was chosen as the explanatory variable.

2) The reasons for choosing the SE-DEA model to measure GUEUL need to be supplemented. The authors should provide the advantages of the SE-DEA model over other methods in measuring GUEUL. Also, the principles of the SE-DEA model should be further explained in the section 2.2.1.

3) The authors should add more information about the internal relationship between each variable and GUEUL to advance the credibility to the selection of the control variables.

4) In the section 4, the authors conduct the further analyses from the perspective of regional socioeconomic and resource endowment, however, it lacks theoretical support.

5) Before explaining the regression results based on the PSM-DID model, the authors should briefly introduce the situation of the GUEUL of the research samples.

Author Response

Point 1: Authors explored the impact of the low-carbon city pilot (LCCP) policy on the green utilization efficiency of urban land (GUEUL). What is the difference between the term “low-carbon” and “green”? Why didn’t study the impact of LCCP policy on low-carbon utilization of urban land? Please explain further in the section “introduction”.

Response 1: we really appreciate this helpful and timely advice. We also found that the two concepts of “low carbon” and “green” are not well explained. Therefore, we add some necessary descriptions to better distinguish the two concepts. Specifically, “green” can accommodate “low carbon”. “Low carbon” is concerned with the level of carbon emissions in urban production and living. In contrast, “green” is a new development model proposed after rethinking and repositioning the relationship between human and nature, and is a rational consideration of the relationship between economic development and environmental ecology, striving to make them coexist and progress together harmoniously. Therefore, compared with studying the impact of low carbon land use in urban areas, we choose to study the efficiency of green land use, which can more comprehensively reflect the harmonious relationship between the change of human-land relationship and economic environment after the implementation of LCCP policy. Thank you again for your suggestions.

Point 2: The reasons why the authors chose SE-DEA model to measure GUEUL need to be supplemented. Compared to other methods, what are the advantages of the SE-DEA model in measuring the GUEUL. Please further explain the principles of the SE-DEA model in the section 2.2.1.

Response 2: Thanks for your suggestion. Frankly, we also found that the elaboration of the reasons for the choice of the SE-DEA model is not detailed. Thus, we re-introduce these parts. On the one hand, we add the advantages of the SE-DEA model compared to the traditional DEA model. In short, the SE-DEA model can compensate for the shortcomings of the DEA model by effectively differentiating and arranging multiple effective DMUs. this model can not only maintain the realism of GUEUL, but also reflect the variability of GUEUL. On the other hand, we add a description of the rationale of the SE-DEA model. Specifically, when performing efficiency analysis, the production frontier of relatively inefficient decision units (efficiency value less than 1) is kept constant and the efficiency value is the same as the calculation result under the traditional model, and the production frontier surface of decision units in the relatively efficient state is recalculated by shifting it backward to obtain a new efficiency value. The SE-DEA model breaks through the limitation of the maximum efficiency value of 1 and can further rank the decision units in the SE-DEA model breaks the limitation that the maximum efficiency value is 1. It can further rank and compare the decision units in the relatively efficient state, and the final efficiency value obtained is the same as that obtained by the traditional DEA model. Thank you again for your suggestion.

Point 3: When choosing control variables, authors should added more information about the internal relationship between each variables and GUEUL.

Response 3: Thanks for your suggestion again. To address this concern, we refer to some literature and add some basic and intuitive explanations. Detailed information can be found in Section 2.3.3.

Point 4:  In the section 4, authors conduct the further analyses from the perspective of regional socioeconomic and resource endowment level. It is lack of explanations and analysis for this chooses, more theoretical explanations should be added.

Response 4: We really appreciate this helpful and timely advice. We also found a lack of explanation and analysis of angle selection. Therefore, we refer to other literature to add some necessary explanations in Section 4. On the one hand, we observe that the unbalanced development of China's regional economies can lead to different effects of the implementation of pilot low-carbon city policies in different regional cities. On the other hand, we argue that different resource-based city types have different energy consumption and industrial structures, and their low-carbon development is a coordinated process of economic, social and ecological systems. Thank you again for your suggestions.

Point 5: Before explaining the regression results based on the PSM-DID model, authors should briefly introduce the situation of the GUEUL of the research samples

Response 5: Thanks for your suggestion. We are also aware of the lack of description of the GUEUL situation. Therefore, in our revision, we have added a brief description of the relevant GUEUL. In we will describe the general GUEUL situation and trends in more detail in section 3.1, which will give the reader a better understanding of GUEUL.

Reviewer 2 Report

The research topic is meaningful and valuable. Still, there are some shortcomings that need to be revised before the paper is officially published.

1) This paper confirmed the positive impact of the LCCP policy on GUEUL, but failed to describe the real situation of GUEUL. I didn’t find the information about the GUEUL in the pilot cities and non-pilot cities. It’s not conducive to an in-depth understanding of the effect of LCCP policy. I think there should be more specific description about the calculation results of GUEUL.

2) Considering the ecological factors in the process of urban land utilization is an innovation of this paper, but the authors failed to show this novelty. The detailed explanation of the index selection of GUEUL should be added.

3) Authors should add the reasons why the Super-efficiency DEA model was used to calculate the GUEUL.

4) The format of the variable names in the Fig.4, Fig.5, and Fig.6 should be consistent with the journal, and the aesthetics of the figure should be improved.

5) The impact of a policy has multiple time-points when using the PSM-DID model. Why is there only one time point in Figure 6? Which year does the term “current” mean?

6) In section 5, the explanation of the impact of the LCCP policy on GUEUL in the central and northeast region is not clear. Authors should added more theoretical analyses so that the subsequent discussion and implications can be more logically and reasonable.

Author Response

Point 1: This paper confirmed the positive impact of the LCCP policy on GUEUL, but failed to describe the real situation of GUEUL. I didn’t find the information about the GUEUL in the pilot cities and non-pilot cities. It’s not conducive to an in-depth understanding of the effect of LCCP policy. I think there should be more specific description about the calculation results of GUEUL.

Response 1: We really appreciate this helpful and timely advice. Honestly, we also find that the description of GUEUL in pilot and non-pilot cities was not specific. Thus, we re-introduce these parts. For the overall GUEUL case, we give a detailed description “We add a description of the specific value change, growth rate, and overall change trend of GUEUL.” In addition, we also describe the information of GUEUL in pilot cities and non-pilot cities in more detail. Figure 2 reflects the information well. We also modify the icon in Figure 2 to make it more in line with the textual description in section 3.1. Thanks for your constructive suggestion again.

Point 2:  Considering the ecological factors in the process of urban land utilization is an innovation of this paper, but the authors failed to show this novelty. The detailed explanation of the index selection of GUEUL should be added.

Response 2: Thanks for your suggestion. In our revised manuscript, we add an introduction to the innovative points of GUEUL's input factor selection. In the new era, the importance of ecological factors in the process of urban land use is gradually increasing. By comparing the efficiency of traditional land use without considering ecological factors, we find that GUEUL can better measure the utilization of excavated land. We provide a more detailed explanation of the GUEUL index selection in section 2.3.1

Point 3: Authors should add the reasons why the Super-efficiency DEA model was used to calculate the GUEUL.

Response 3: Thanks for your suggestion. Frankly, we also found that the elaboration of the reasons for the choice of the SE-DEA model is not detailed. Thus, we re-introduce these parts. On the one hand, we add the advantages of the SE-DEA model compared to the traditional DEA model. In short, the SE-DEA model can compensate for the shortcomings of the DEA model by effectively differentiating and arranging multiple effective DMUs. this model can not only maintain the realism of GUEUL, but also reflect the variability of GUEUL. On the other hand, we add a description of the rationale of the SE-DEA model. Specifically, when performing efficiency analysis, the production frontier of relatively inefficient decision units (efficiency value less than 1) is kept constant and the efficiency value is the same as the calculation result under the traditional model, and the production frontier surface of decision units in the relatively efficient state is recalculated by shifting it backward to obtain a new efficiency value. The SE-DEA model breaks through the limitation of the maximum efficiency value of 1 and can further rank the decision units in the SE-DEA model breaks the limitation that the maximum efficiency value is 1. It can further rank and compare the decision units in the relatively efficient state, and the final efficiency value obtained is the same as that obtained by the traditional DEA model. Thank you again for your suggestion.

Point 4: The format of the variable names in the Fig.4, Fig.5, and Fig.6 should be consistent with the journal, and the aesthetics of the figure should be improved.

Response 4: Thanks for your suggestion. We redrew Fig.4, fig.5, and Fig.6 to address this issue by bringing them into compliance with the article's content and aesthetic standards. In addition, in order to make Fig.3's legend consistent with other images, we also change it. Detailed information can be found in Fig.3, Fig.4, Fig.5, and Fig.6.

Point 5: The impact of a policy has multiple time-points when using the PSM-DID model. Why is there only one time point in Figure 6? Which year does the term “current” mean?

Response 5: Thanks for this helpful suggestion. We also find a lack of a good explanation for the shock time setting in Fig. 6. Due to the time of 2010 is the first batch of pilot, we choose 2010 as the "current". If we choose the three pilot periods for dynamic testing at the same time, the pre-pilot parameters of the second and third batches will be affected by the pilot policies of the first batch. From the principle of dynamic effect test, we test that there is no difference in the change trend of GUEUL between pilot cities and non-pilot cities before 2010, which satisfies the common trend hypothesis required by the principle. Thanks for your constructive suggestion again.

Comment 6: In section 5, the explanation of the impact of the LCCP policy on GUEUL in the central and northeast region is not clear. Authors should added more theoretical analyses so that the subsequent discussion and implications can be more logically and reasonable.

Response 6: Thanks for this helpful suggestion. We also found a lack of detailed theoretical analysis in the central and northeastern regions. In Section 5, we offer details on the practical considerations and development challenges encountered by the central and northeastern regions in constructing low-carbon cities by citing several academic works. Based on this basis, we offer a more logical analysis of how low-carbon urban policies affect GUEUL in the two regions. Thanks for your constructive suggestion again.

Round 2

Reviewer 1 Report

The authors have addressed most of my concerns, and I recommend it to be accepted for publication.